# Dyspnea Severity Assessment Based on Vocalization Behavior with Deep Learning on the Telephone

**DOI:** 10.3390/s23052441

**Published:** 2023-02-22

**Authors:** Eduardo Alvarado, Nicolás Grágeda, Alejandro Luzanto, Rodrigo Mahu, Jorge Wuth, Laura Mendoza, Néstor Becerra Yoma

**Affiliations:** 1Speech Processing and Transmission Laboratory, Electrical Engineering Department, University of Chile, Santiago 8370451, Chile; 2Clinical Hospital, University of Chile, Santiago 8380420, Chile

**Keywords:** respiratory distress estimation, deep learning, telephone speech

## Abstract

In this paper, a system to assess dyspnea with the mMRC scale, on the phone, via deep learning, is proposed. The method is based on modeling the spontaneous behavior of subjects while pronouncing controlled phonetization. These vocalizations were designed, or chosen, to deal with the stationary noise suppression of cellular handsets, to provoke different rates of exhaled air, and to stimulate different levels of fluency. Time-independent and time-dependent engineered features were proposed and selected, and a k-fold scheme with double validation was adopted to select the models with the greatest potential for generalization. Moreover, score fusion methods were also investigated to optimize the complementarity of the controlled phonetizations and features that were engineered and selected. The results reported here were obtained from 104 participants, where 34 corresponded to healthy individuals and 70 were patients with respiratory conditions. The subjects’ vocalizations were recorded with a telephone call (i.e., with an IVR server). The system provided an accuracy of 59% (i.e., estimating the correct mMRC), a root mean square error equal to 0.98, false positive rate of 6%, false negative rate of 11%, and an area under the ROC curve equal to 0.97. Finally, a prototype was developed and implemented, with an ASR-based automatic segmentation scheme, to estimate dyspnea on line.

## 1. Introduction

Chronic respiratory diseases (CRDs) generate a high burden on healthcare systems around the world [1]. It is estimated that 262 million people suffer from bronchial asthma, and more than 200 million people suffer from chronic obstructive pulmonary disease (COPD), making them the most common CRDs recorded. In 2019, more than 3 million people died from COPD, which accounted for 6% of deaths that year worldwide [2], and it is expected to be the third leading cause of mortality globally by 2030 [3,4]. Although CRDs are not curable, the treatment of these diseases allows better control of their symptoms, which therefore improves the quality of life of people that suffer from them [5].

One of the most widely employed methods to detect and monitor respiratory conditions is by X-ray, due to its speed, accessibility, and low cost. Another widely utilized method is CT imaging, which allows for visualization and quantitative detection of disease severity [6]. Biomedical experts also employ the analysis of sounds generated by the respiratory system (lung noise, coughing, breathing, voice, and hearbeats) to detect respiratory conditions such as asthma, bronchitis, pertussis, and SARS-CoV-2 [7]. The spirometry test can also detect pulmonary disorders, but its result must be carefully interpreted by a medical specialist [8]. Notably, one thing that is repeated in all the previous methods is that the patient must go to a clinical interview or to the health center to undergo the corresponding test, which in turn involves natural restrictions regarding the patient´s age and location.

In recent years, there has been a considerable increase in the development of remote monitoring health tools, due to the growing demand for health services [9]. There has also been a great effort in trying to prevent COPD with the use of machine learning (ML), since these methods are effective in collecting and integrating diverse medical data on a large scale, in precision medicine [10]. This development has been boosted by the COVID-19 pandemic [11], whereby several artificial intelligence (AI)-based solutions have been proposed to automatically detect SARS-CoV-2 [12,13,14,15]. These ML-based solutions have been focused mainly on smart phones due to the great scalability, ubiquity, and flexibility that these devices offer [16]. Usually, these systems are useful for disease monitoring and preventing people from having to visit medical centers. For example, in [17], the respiration rate was remotely evaluated by using phone sensors. In [18], the importance of telemedicine for people with COPD was evaluated, and in [19], the advantages of remote monitoring for patients with interstitial lung diseases were studied. The relevance of remote monitoring for people with chronic critical illness, who have already been discharged, is discussed in [20].

In addition to the importance of remote monitoring, automation allows greater scalability, by not requiring a specialist to evaluate each person, and several automatic health applications have been proposed in the last few years. For example, the automatic detection of COVID-19 using X-ray and CT images of the lungs as input, has been extensively addressed [21]. Other studies have focused on the automatic detection of SARS-CoV-2 by means of analyzing the audio generated by forced coughing, vocalizations, and breathing [22,23].

With regards to [24], a review of different studies, methods, and databases that focused on the remote monitoring of respiratory diseases through audio analyses was carried out. Detection by means of audio analysis makes it possible to standardize the evaluation; reducing variability or bias between different doctors, that took the test in the form of a questionnaire. In particular, although the dyspnea assessment questionnaire is easy for a doctor to apply, it is not for ordinary people, who may have complications in understanding or answering the questions, particularly in the case of elderly adults. Moreover, the responses given in a questionnaire can be influenced by the patient’s mood or habituation to the disease [25,26,27].

Coughing is a common symptom of both colds and respiratory conditions, which accounts for about 38% of the respiratory disorder requests [28]. Despite the fact that it can be considered an important source of information for machine learning (ML)-based schemes, prompting the user to repeat coughing events compromises the naturalness of the symptom and can be a source of discomfort. Some researchers, however, claim that coughing is not the most reliable symptom to identify respiratory diseases such as COVID-19, and that it obtained worse results compared to vocalizations such as a sustained vowel or text reading [29].

As mentioned already, studies on the automatic identification of respiratory conditions have focused mostly on COVID-19 [30], but have also included diseases such as asthma, bronchitis, and pertussis [31]. However, the severity of the respiratory symptoms has hardly been addressed. In fact, the degree of symptom severity is a very important metric for monitoring patients, as well as for a first diagnosis. One exception is presented in [32], where a method is proposed to classify patients in different degrees of COPD on a scale from one (mild) to four (very severe) according to FEV1 (forced expiratory volume).

Open-source databases are used to train ML models such as COSWARA [33], DICOVA [34], or COUGHVID [35], among others, as well as private databases, which show some similarities in the recorded audio, such as the use of sustained vowels, breathing, sentence readings, or forced coughing. Despite the fact that the use of sustained vowels is quite common, it is important to bear in mind that the noise suppression schemes of cell phones may attenuate stationary signals. Studies such as [36] employ the same microphones for all the participants, to avoid any audio preprocessing mismatch. Moreover, public or private databases are usually small, because they are difficult to produce, which in turn requires optimization of the training procedure to maximize the final accuracy and robustness. Data augmentation methods such as time shifts [30] and k-fold cross-validation training [29], are frequently adopted to increase the number of training examples.

ML-based schemes that employ speech as input usually extract features such as Mel-frequency cepstral coefficients (MFCCs) and Mel-frequency spectrograms, which have been widely employed in automatic speech recognition (ASR) [37], and have also been proposed in [38] for respiratory distress detection. Furthermore, the first and second derivatives of these coefficients allow for the evaluation of the dynamics of the voice signal [39]. Other features such as pitch, jitter, and shimmer were proposed in studies such as [40] and for COVID-19 detection.

The optimization of ML architectures and parameters is a common practice, as can be seen in [30,41,42,43,44], where the problems of COVID-19 or respiratory distress detection were addressed by employing convolutional neural network (CNN) layers to obtain deep features. The resulting features were concatenated and input to a neural-network-based classifier that was trained on an end-to-end basis to combine the parameters. Staged training has also been adopted: first, classification modules are trained independently with each set of features; then, the output of the classifiers is combined to obtain the final system decision. This kind of strategy allows the optimization of the information delivered by each set of features and the exploration of classification fusion methods, which is not possible with single neural network architecture. For instance, in [26], the outputs of the classification modules (i.e., softmax) are input to an SVM to obtain the final decision. In [45], the final decision is obtained by applying the majority vote rule to the classifier outputs. In [46], the output probabilities are weighted to obtain the final classification decision.

Surprisingly, the optimization of the complementarity that can be provided by different types of phonetizations has not been addressed exhaustively. In some cases, as in [29], the VGG19 CNN architecture was employed to find the vocalization that could provide the highest accuracy in post COVID-19 patient identification. In other studies, such as in [31], the features extracted from the phonetizations are concatenated and input to a neural network that is expected to learn how to combine them.

This paper presents a system that detects dyspnea automatically over the telephone. This design allows monitoring of the breathlessness status of patients, ubiquitously and remotely, with the modified Medical Research Council (mMRC) scale. The mMRC allows the classification of respiratory distress in five levels, from zero (healthy) to five (very severe). Surprisingly, this topic has not been addressed exhaustively in the literature, but most related studies have focused on the binary detection of COVID-19 or respiratory conditions.

The database used to train the system consists of three controlled vocalizations after taking deep breaths and until gasping for air, which were designed to represent the user’s behavior while performing them. The first two phonetizations correspond to/ae-ae/and/sa-sa/, and provide relevant information about the amount of air exhaled by the individuals. In contrast to sustained vowels employed elsewhere, they were not stationary and are not cancelled by the noise suppression scheme in smart phones. The third phonetization corresponds to counting from one to thirty as fast as possible, to evaluate the spontaneous behavior of the subjects who must make an effort to reach the goal. The motivation is to cause involuntary breathing, voice pauses, coughing, tone variation, etc., that could characterize dyspnea severity.

The proposed method extracts time-dependent and time-independent features from each phonetization. Thereafter, an individual classifier is trained per each kind of feature and phonetization. By doing so, the dimensionality of the input vector of the models can be reduced. Additionally, this strategy provides more degrees of freedom to make use of the complementarity resulting from the different vocalizations and the information that can be obtained from them. This issue has not been tackled in depth in the literature either.

The classifiers employed for the time-independent features corresponded to multilayer perceptrons (MLP). For the time-dependent features, architectures based on CNN are employed for the /ae-ae/ and /sa-sa/ vocalizations. In the case of the one-to-thirty counting, an architecture based on CNN and the long short-term memory (LSTM) neural network was adopted.

The results obtained in this paper show that dyspnea can be detected and estimated with an accuracy of 59% (i.e., the same mMRC score), with a root-mean-square error (RMSE) equal to 0.98. The obtained false positive (FP) and false negative (FN) rates were 6% and 11%, respectively. The area under the ROC curve (AUC) was equal to 0.97. The main contributions of this paper concern: automatic assessment of dyspnea on the phone using a severity scale, i.e., mMRC; modelling the spontaneous behavior of individuals when prompted to produce controlled phonetizations, where behavior includes the pitch variation, speed changes, involuntary pauses or voice breaks, involuntary coughing, etc.; vocalization selection that takes into consideration the noise suppression of cellular devices, the air volume exhaled, and phonetic variability; a method to combine the information provided by different types of features and phonetizations; and, a k-fold-based training system with two validation sets. It is important to emphasize that the assessment of dyspnea severity with the method proposed here goes beyond the COVID-19 pandemic. For example, it can be used in telemedicine, the monitoring of seasonal bronchopulmonary diseases, the effect of contamination in slaughter areas, and occupational diseases.

## 2. Methodology

### 2.1. Dataset

The database is composed of patients with respiratory conditions (COPD, pulmonary fibrosis, COVID-19) recruited at the Clinical Hospital at University of Chile (HCUCH, *Hospital Clínico de la Universidad de Chile*), and healthy volunteers from the Faculty of Physical and Mathematical Sciences (FCFM, *Facultad de Ciencias Físicas y Matemáticas*) at the same university. The study was approved by the scientific ethics committees at the HCUCH and the FCFM. Those who were included in the database had to give informed consent to participate in the study. They were thereafter interviewed by a pulmonologist at HCUCH, who evaluated the degree of dyspnea using the mMRC scale (gold standard). Each participant´s mMRC score was used as a target for training the system.

The voice recordings employed here consist of three types of vocalizations that the individuals were prompted to produce, without pauses, after taking deep breaths, and until they gasped for air. These were, the sequence of Spanish phonemes /a/ and /e/, denoted here as /ae-ae/; the sequence of Spanish syllables /sa/, indicated here as /sa-sa/; and, the last one was inspired by the Roth test [47], where the subjects were asked to count in Spanish from one to thirty, or until they gasped for air as fast as they could. From the /ae-ae/ vocalization, information very similar to that of a sustained vowel (continuous sequence) was obtained, but it avoided the problem of attenuation caused by the noise suppression scheme of smart phones, because the corresponding speech signal is less stationary than a single sustained vowel, such as /a/, for example. The sequence /sa-sa/ is not stationary either, it must be repeated as fast as possible, and the exhalation rate of air volume is higher than in the case of /ae-ae/, because the vocal folds are distended when the voiceless phoneme /s/ is being produced, in contrast to voiced phonemes such as /ae-ae/. The telephone channel cut off frequency of 4 KHz dramatically reduces the sample amplitude of /s/, but the vowel /a/ in the sequence /sa-sa/ allows for the detection of the corresponding signal. The speech resulting from counting from one-to-thirty is highly non-stationary, and allows a better representation of the user’s speaking behavior, such as pauses, intonation changes, speaking speed, etc., while uttering continuous speech. Interestingly, it was observed that these controlled vocalizations avoided forced situations or behaviors such as coughing.

The database was composed of 104 participants, where 34 corresponded to healthy individuals and 70 were patients with respiratory conditions (44 COPD, 21 pulmonary fibrosis, and five sequelae of COVID-19). An mMRC score equal to zero was allocated to the healthy participants. The patients were clinically evaluated with respect to their mMRC score, resulting in 19 with an mMRC score equal to 1; 29 with an mMRC score equal to 2; 20 with an mMRC score equal to 3; and two with an mMRC score equal to 4. These scores, that were obtained by means of clinical evaluation, were employed as references, or the gold standard, to train our deep-learning-based models. The models were trained with k-fold cross-validation, where two validation subsets per partition were adopted. Conventionally, the first subset had the purpose of stopping the learning procedure before overfitting. However, the second one was employed to analyze the generalization capability of the models, obtained by repeating the training process several times. The number of patients with mMRC = 4 was too low (i.e., two) resulting in an underrepresented class. Consequently, these subjects were incorporated to the subset of individuals with mMRC = 3, leading to four classes, with the mMRC score ranging from 0 to 3, where level 3 corresponded to the most severe breathlessness condition in our case.

Subsequently, in order to obtain the recordings of the phonetizations, people were contacted by telephone with an IVR system. The individuals were prompted to repeat each vocalization twice, following the procedure aforementioned. The audios obtained were stored in WAV format, with a sampling rate of 8 kHz, and were assigned a random ID to protect the identity of the participants. The database was composed of 104 people, so each phonetization had 208 audios (two repetitions per individual), and the total dataset reaches 624 vocalizations. After receiving all the audios, an automatic speech recognition (ASR) system was trained to isolate the target vocalizations from the background noise or undesirable audio.

### 2.2. The Proposed Method

The system aimed to characterize the behavior of the users when performing controlled phonetization, to classify their dyspnea level on the mMRC scale. As discussed above, the controlled vocalizations were chosen to provide some degree of complementarity between them, and to counteract the noise suppression scheme of smartphones. The selected phonetizations allow representation of the user´s phonetic articulation spontaneous behavior such as pauses, intonation variation, vocalization length, speaking speed, and non-voluntary coughing, breathing, or pauses. Notably, in order to realize these, time-dependent and time-independent features were defined, and extracted independently from the speech signals. The time-dependent features were computed on a frame-by-frame basis, and attempted to capture the dynamics of the vocalization signals to represent pauses, speaking speed, and non-voluntary coughing or breathing. They correspond to: /ae-ae/ and /sa-sa/ phonetizations, Mel filters estimated from the FFT log power spectrum; and, one-to-thirty counting, FFT log power spectrum. On the other hand, time-independent features aimed to characterize the whole vocalization signals, by providing information such as the phonetization length, and intonation curve variation and slope.

Although the features were carefully chosen or designed, deep learning schemes were necessary to obtain the final dyspnea mMRC score. One of the contributions of the proposed approach is the fact that it does not require situations or behaviors to be forced unnaturally, such as non-spontaneous coughing. In contrast, it relies on phonetizations that can easily be replicated more naturally. As time-dependent and time-independent features characterize users’ behavior with complementary representations, combining them should result in a more accurate and robust final classifier.

The classical classification loses the ordinality of the labels, since it considers these as independent [48]. However, the use of regression also suffers from the problem that the root mean square error assumes that the separation between adjacent levels of the mMRC scale would be uniform [49]. In fact, although not reported here, the regression performed worse overall than the classification-based system. It is important to emphasize that the regression restricts the flexibility in merging or combining the different modules to explore the complementarity of their outputs. For these reasons, neuron stick-breaking [50] was considered as a trade-off between both solutions, including ordinality in the classification problem. The stick-breaking layer provided better results in some cases.

Figure 1 shows the block diagram of the system presented here. Each type of vocalization provided a four dimension softmax, representing the probability of each mMRC score. These three phonetization dependent softmax were combined with the following five rules, generating five new softmax: minimum, maximum, mean, median, and product. These five outputs were averaged to generate the final softmax, where the estimated mMRC score corresponds with the highest probability.

Figure 2 shows how the vocalization dependent scores were obtained. There were two classifiers per type of phonetization, one that received the time-dependent features and another one for the time-independent parameters. Each type of vocalization was repeated twice by the individuals. After extracting the time-dependent and time-independent features, they were propagated through the corresponding machine learning module that outputs softmax per each repetition and kind of parameter. The time-dependent features employed a CNN- or LSTM-based architecture, and the time-independent parameters made use of an MLP scheme. The resulting time-dependent and independent softmax delivered by each repetition were combined separately, using the same scheme described above (Figure 1), with five combination rules, to obtain a single softmax per feature type. Thereafter, the time-dependent and -independent feature softmax were combined by a simple average to deliver the vocalization dependent softmax. This process was replicated for each type of vocalization to deliver the estimated mMRC score, as shown in Figure 1.

#### 2.2.1. MLP and Time-Independent Features

Two of the time-independent features were computed with the fundamental frequency, F_0_, estimated on a frame-by frame basis with Praat [51]. To represent the subjects´ behavior with respect to the F_0_ curve [52], the following features were extracted within each vocalization: the mean of the normalized slope and the standard deviation. The third parameter corresponds to the phonetization length in seconds. The mean and variance normalization (MVN) was applied to each parameter, where the mean and variance of each parameter were computed within the whole database. As stated above, one time-independent feature, MLP, was trained per each type of phonetization, i.e., /ae-ae/, /sa-sa/, and one-to-thirty counting (see Figure 2). The learning rate was made equal to 0.001, ADAM optimizer and cross entropy as a loss function was employed. The hidden layers employed the ReLU activation function. The output layer had four neurons with softmax activation. In the case of /ae-ae/, the network had two hidden layers of 20 neurons each. The MLP corresponding to /sa-sa/ used one 20-node hidden layer. Finally, the one-to-thirty counting vocalization made use of a network with one 30-node hidden layer.

#### 2.2.2. Neural Network Architectures for Time-Dependent Features

The time-dependent features were based on the FFT log power spectrum, and were optimized for each type of phonetization. The 512-sample FFT was estimated in 50 ms windows with 50% overlap, where 257 frequency bins were obtained. Fourteen Mel filter log energies/frame were thereafter computed in the case of the /ae-ae/ and /sa-sa/ phonetizations. In the case of the one-to-thirty counting vocalization, Mel filters were not employed, but 75% of the lower frequency bins of the log spectrum was selected, and the corresponding first derivative or delta features were included, resulting in 257 × 0.75 × 2 = 386 features/frame. MVN is applied on the time trajectories of the time-dependent features where the parameter means and variances are computed on the whole database. Finally, zero padding was performed based on the longest utterance in the training data, corresponding to the same type of phonetization. The time-dependent feature architecture and hyperparameter optimization led to: the use of neuron stick breaking, a learning rate equal to 0.0001, and ADAM optimizer and cross entropy as the loss function. The resulting deep learning architectures are shown in Figure 3a (/ae-ae/ and /sa-sa/ vocalizations) and Figure 3b (one-to-thirty counting).

#### 2.2.3. K-Fold Training with Double Validation

To optimize the available database, a nine-fold cross-validation was performed. Twelve users from the database were extracted in the first five partitions, and 11 people in the remaining four partitions, for testing. It is important to mention that this data division scheme ensures that a given speaker could not have vocalizations in the training, validation, or testing subsets simultaneously. Besides testing the individuals, each partition was composed of training, validation 1 and validation 2 subsets, corresponding to 70%, 15%, and 15% of the partition individuals, respectively.

The classifiers were trained eight times with each partition, to take into consideration the variability due to weight initialization. The training subset was used to estimate the network weights, and validation 1 data was employed to stop the iterations and avoid overfitting, with an early stopping of 20. For each partition, the optimal neural network classifier was chosen, among the eight that were trained, by picking the one with the highest average accuracy evaluated on the validation 1 and validation 2 subsets. The latter did not make part of the training procedure, so the chosen trained neural network is also the one with the best generalization capability. Test data, which was never seen by the network, was propagated to obtain the mMRC scores and metrics for the corresponding partition. These steps were replicated for all the partitions to obtain the scores and metrics for all the 104 individuals. Finally, the whole procedure was repeated five times to obtain more reliable statistics.

### 2.3. Performance Metric

The metrics adopted to evaluate the system performance were: mMRC score accuracy; root mean square error (RMSE); false positive rate (FP); false negative rate (FN); and, area under the ROC curve (AUC). RMSE was calculated as follows:$$\mathrm{RMSE}=\sqrt{\frac{\sum_{i=1}^{N}{\left({\mathrm{Estimated}\;\mathrm{mMRC}\;\mathrm{score}}_{i}-{\mathrm{Reference}\;\mathrm{mMRC}\;\mathrm{score}}_{i}\right)}^{2}}{N}}$$
where ${\mathrm{Estimated}\;\mathrm{mMRC}\;\mathrm{score}}_{i}$ and ${\mathrm{Reference}\;\mathrm{mMRC}\;\mathrm{score}}_{i}$ denote the estimated and reference mMRC scores assigned to user i, and N is the total number of individuals in the database. A false positive event is defined as an individual whose reference mMRC score is equal to 0, but who was assigned an estimated mMRC score equal to one, two, or three. In contrast, a false negative event is defined as an individual whose reference mMRC score is greater than or equal to one, but was assigned an estimated mMRC score equal to zero. Although the classification is carried out with four classes (i.e., mMRC from zero to three), metrics such as FP, FN, and AUC, are obtained on a binary basis, where class 0 corresponds to the healthy condition, and an mMRC score from 1 to 3 indicates the presence of dyspnea.

### 2.4. Feature, Architecture, Hyperparameter, and Training Optimization

The time-independent features that were considered initially were duration, average pitch, pitch slope, pitch standard deviation, jitter, voice breaks, and, energy center per frame. Subsequently, the features that provided the highest discrimination between individuals with (i.e., reference mMRC score equal to 1, 2, or 3) and without (i.e., reference mMRC score equal to 0) dyspnea were chosen: the pitch slope normalized by the average F_0_; the standard deviation of the F_0_ curve; and the vocalization duration in seconds. The time-independent feature fully connected MLP was tuned with respect to: learning rates, i.e., 0.1, 0.01, 0.001, or 0.0001; number of neurons per layer, i.e., 10, 20, 30, 40, 50, or 60; the number of hidden layers, i.e., 1, 2, 3, 4, or 5; and the use of neuron stick-breaking.

In the case of the time-dependent features, the following configurations were tested: number of FFT samples, i.e., 128, 256, 512, and 1024; window length, i.e., 128, 256, and 512 samples; bandwidth from the first FFT bin, i.e., 25%, 50%, 75%, and 100% of the FFT bins; FFT log spectrum vs. Mel filter log energy; and, with or without delta and delta–delta features. The window overlap was made equal to 50% and the number of Mel filters was 14. Regarding the time-dependent feature neural networks, a more exhaustive optimizations was carried out: 1D CNN convolutional networks, i.e., kernel size (3, 5, or 7), numbers of filters (16, 32, 64, or 128), and number of convolutional layers (3, 6, 10, or 14); max pooling blocks; residual connections; LSTM or BiLSTM, i.e., number of layers and dimensionality; as well as the use of neuron stick-breaking. As depicted in Figure 3a, two convolutional layers plus max pooling was considered as a single block, which in turn was replicated a number of times that it was tuned. A final output, fully connected block, was also optimized by tuning: number of layers, i.e., one, two, or three; and, number of neurons per layer, i.e., 16, 32, 64, 128, or 256. The output of the fully connected block was composed of four softmax nodes, corresponding to the four mMRC scores or classes.

### 2.5. Implementation of Telephone and Web Application

A platform was designed and implemented to record the database. It was used to test the proposed system in real-time. Figure 4 shows the deployed infrastructure, where users can record their vocalizations with a telephone call (i.e., with a IVR server) or using the phone with a web-based application, despite the fact that the results presented here were obtained with speech data recorded with the IVR server only. The audio recorded by each individual is stored in the cloud, and notifies the web service when it is available. A daemon process checks if there is audio to be processed. If so, it is downloaded and segmented with ASR technology to remove background noise or spurious signals, such as other people´s speech, audio from TV or radio sets, etc. After ASR segmentation, the features are extracted and processed with the neural-network-based systems to deliver the estimated mMRC score, which in turn is returned to the user by making use of the IVR or web-based application servers. In the database base recording mode, no mMRC score is returned to the user.

## 3. Results and Discussion

According to Figure 5a, where the results with time-independent features are presented, the highest accuracy (blue bars) corresponds to the combination of the classifiers provided by the three types of phonetizations, i.e., 53%. The worst accuracy is delivered by the one-to-thirty counting, i.e., 38%. This outcome is corroborated when the /ae-ae/ and /sa-sa/ classifiers are combined, giving an accuracy almost as high as the best one, i.e., 52%. This result may suggest that the time-independent features give a slight increase in accuracy, reaching a maximum of 53% when using the three types of phonetizations. Note, the combination of classifier outputs will be denoted with ⊕. The lowest RMSE’s (red bars) are obtained with /ae-ae/⊕/sa-sa/⊕one-to-thirty counting or with/ae-ae/⊕one-to-thirty counting. Moreover, the time-dependent features provided a lower score dispersion depending on the phonetization classifier fusions, but the highest accuracy and the lowest RMSE also occurred when the scores from the three phonetizations were combined, as depicted in Figure 5b, i.e., /ae-ae/⊕/sa-sa/⊕one-to-thirty counting. When the output of the time-independent and time-dependent classifiers are combined (Figure 5c), the highest accuracies took place with /ae-ae/⊕/sa-sa/ and with /ae-ae/⊕/sa-sa/⊕one-to-thirty counting, which are 16% and 12% higher, respectively, than those obtained with time-independent or time-dependent features using the same classifier combination. Similarly, the lowest RMSE occurred with/ae-ae/⊕/sa-sa/⊕one-to-thirty counting, which in turn is 9% and 11% lower, than those obtained with time-independent or time-dependent features, respectively, using the same classifier combination.

Figure 6a–c depicts FP and FN with time-independent and time-dependent features, and the combination of both types of parameters, respectively. As can be seen in Figure 6a–c, the score fusion provided by more than one phonetization usually gave lower FP and FN than single vocalizations. In addition, the lowest average FN + FP across the three subplots took place with the fusion of the three types of phonetization, which were 36%, 37%, 42%, 15%, 10%, and 23% lower than the average FN + FP with /ae-ae/, /sa-sa/, one-to-thirty counting, /ae-ae/⊕/sa-sa/, /ae-ae/⊕one-to-thirty counting, and/sa-sa/⊕one-to-thirty counting, respectively. Moreover, when time-dependent and time-independent features are used in Figure 6c, average FN+FP with /ae-ae/⊕/sa-sa/⊕one-to-thirty counting were 61%, 58%, 59%, 36%, 26%, and 42% lower than FN+FP obtained with /ae-ae/, /sa-sa/, one-to-thirty counting, /ae-ae/⊕/sa-sa/, /ae-ae/⊕one-to-thirty counting, and/sa-sa/⊕one-to-thirty counting, respectively.

Figure 7 shows AUC with the different combinations of vocalizations, when using both the time-dependent and time-independent features. The results with AUC are similar to those in Figure 5 and Figure 6. The best results were achieved when using the combination of phonetization classifiers. The combination of the three vocalizations gave the highest AUC, which were 10%, 6%, 9%, 3%, 2%, and 2% higher than those obtained with /ae-ae/, /sa-sa/, one-to-thirty counting, /ae-ae/⊕/sa-sa/, /ae-ae/⊕one-to-thirty counting, and/sa-sa/⊕one-to-thirty counting, respectively.

## 4. Conclusions

This paper proposed a system to assess dyspnea with the mMRC scale on the phone, by making use of deep learning. The method models the spontaneous behavior of subjects while pronouncing controlled vocalizations, which in turn were designed or chosen to cope with the stationary noise suppression of cellular handsets leading to different rates of exhaled air, and to motivate different levels of fluency. Time-independent and time-dependent engineered features were proposed and tested, and a k-fold scheme with double validation was employed to pick the models with the highest potential generalization capabilities. Moreover, score fusion methods were also explored, to optimize the complementarity of the three types of controlled phonetizations and the features that were designed and selected. The database was composed of 104 participants, where 34 corresponded to healthy individuals and 70 were patients with respiratory conditions (44 COPD, 21 pulmonary fibrosis, and five sequelae of COVID-19). The results presented here were obtained with the subjects´ vocalizations that were recorded with telephone calls (i.e., with the IVR server). Moreover, a prototype was developed and implemented with an ASR-based automatic segmentation scheme, to estimate dyspnea online.

The system provided an accuracy of 59% (i.e., estimating the correct mMRC), a root mean square error equal to 0.98, false positive rate of 6%, false negative rate of 11%, and an area under the ROC curve equal to 0.97. These results are in the range of the accuracies of clinical tests, which suggests that the technology presented here is a candidate to be deployed in public health applications, and can detect dyspnea automatically by making use of the telephone network with artificial intelligence, without any prior knowledge or tests on subjects. The application of this technology could thus help to monitor the population at risk of pneumonia by COVID-19, and to detect COVID-19 sequelae. Moreover, it offers the opportunity to have a remote and reliable tool beyond the current pandemic. For example, it could be employed for dyspnea screening in the general population, allowing the opportunity of diagnosis and management of bronchopulmonary illnesses. Interestingly, it could be helpful to monitor respiratory diseases, to evaluate the effect of pollution, to monitor pre-existing or occupational diseases such as byssinosis in bakers, or those resulting from high-risk pulmonary tasks such as work in large-scale mining or firefighting with dangerous gasses, amongst other occupations with high respiratory compromise.

Patients with severe respiratory distress should have no problem performing controlled phonetizations. In cases where the health status of the patient is so severe to the point that it does not allow them to pronounce these controlled vocalizations (e.g., when connected to an artificial respirator), the proposed system is clearly not applicable. Nonetheless it could be used to detect worsening respiratory distress before reaching an acute phase. A weakness of this study corresponds to the subjacent hypothesis, that the manifestation of dyspnea does not depend on the illness that causes it. Discriminating dyspnea depending on its cause is out of the scope of this paper. However, the aforementioned hypothesis seems reasonable. The aim of the technology presented here was to detect dyspnea independently of the underlying cause. Nevertheless, to determine if there are differences in observed dyspnea with respect to gender, age, comorbidities, etc., can be proposed as future research.

## Figures and Tables

**Figure 1 sensors-23-02441-f001:**
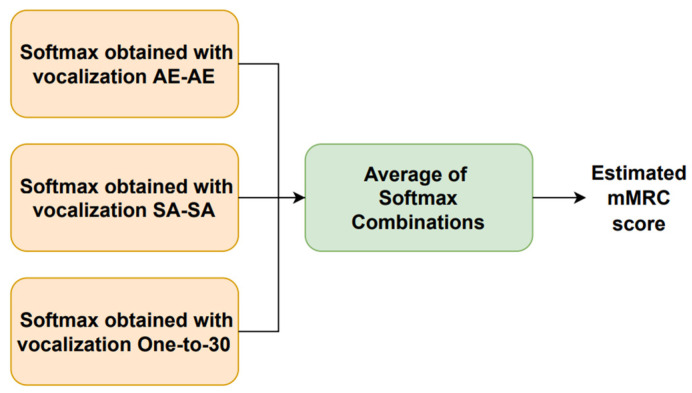
Block diagram of the proposed system.

**Figure 2 sensors-23-02441-f002:**
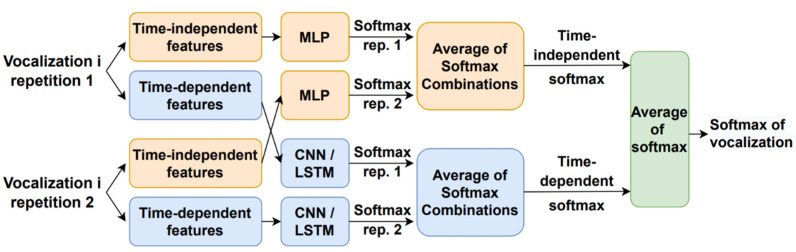
Estimation of the vocalization dependent scores.

**Figure 3 sensors-23-02441-f003:**
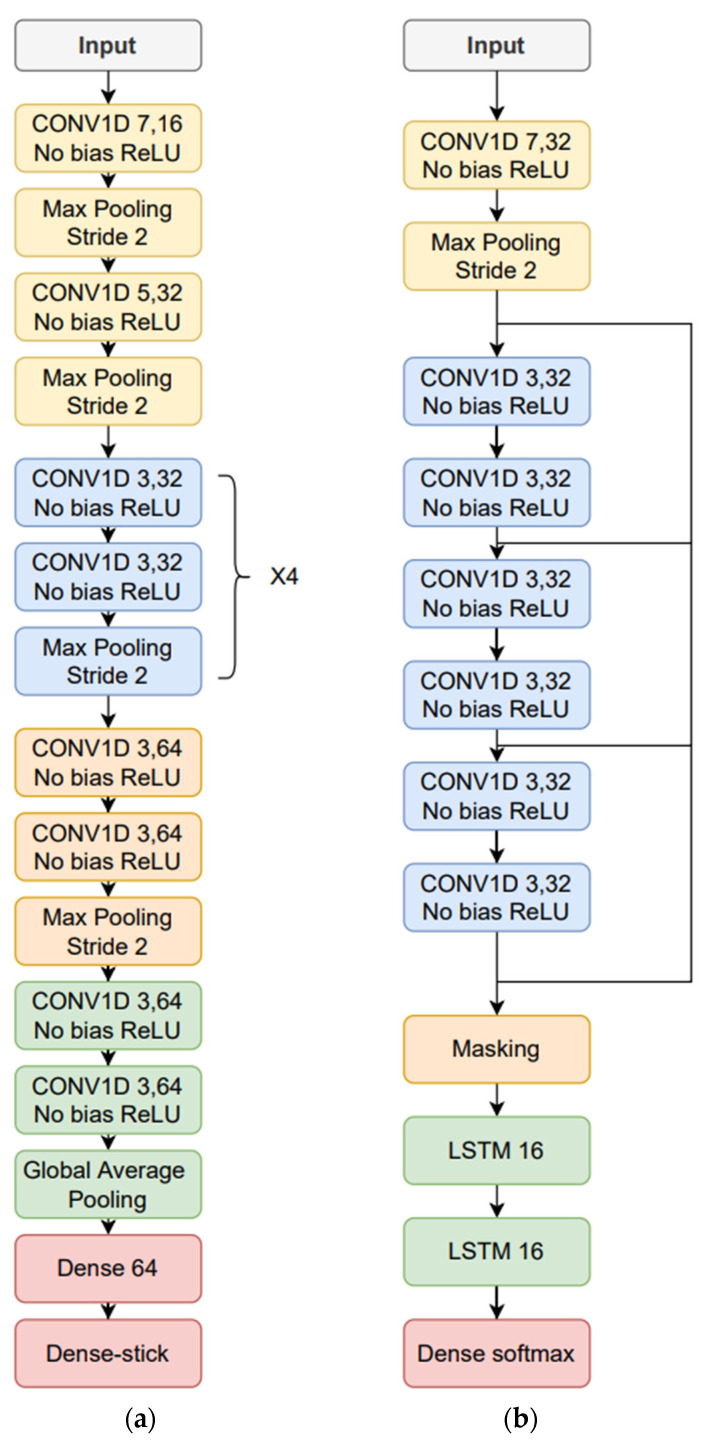
Neural network architectures for: (**a**) /ae-ae/ and /sa-sa/ phonetization; and, (**b**) one-to-thirty counting vocalization.

**Figure 4 sensors-23-02441-f004:**
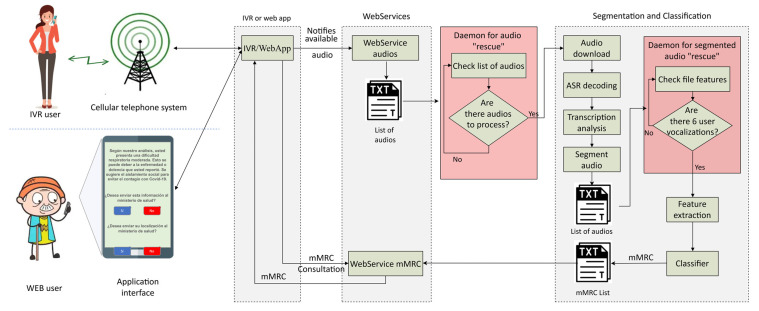
IVR and web-based application platform.

**Figure 5 sensors-23-02441-f005:**
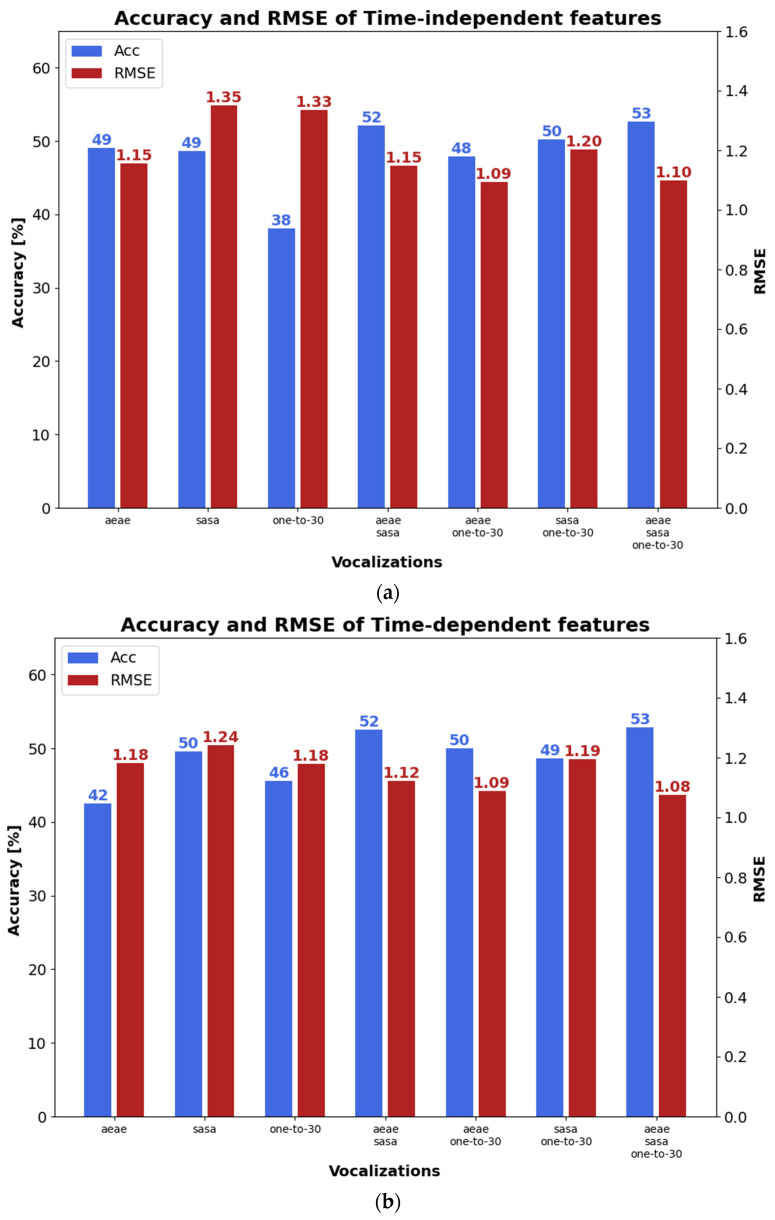
Accuracy and RMSE with single and combined classifiers: (**a**) time-independent features; (**b**) time-dependent features; and, (**c**) combined time-independent and time-dependent features.

**Figure 6 sensors-23-02441-f006:**
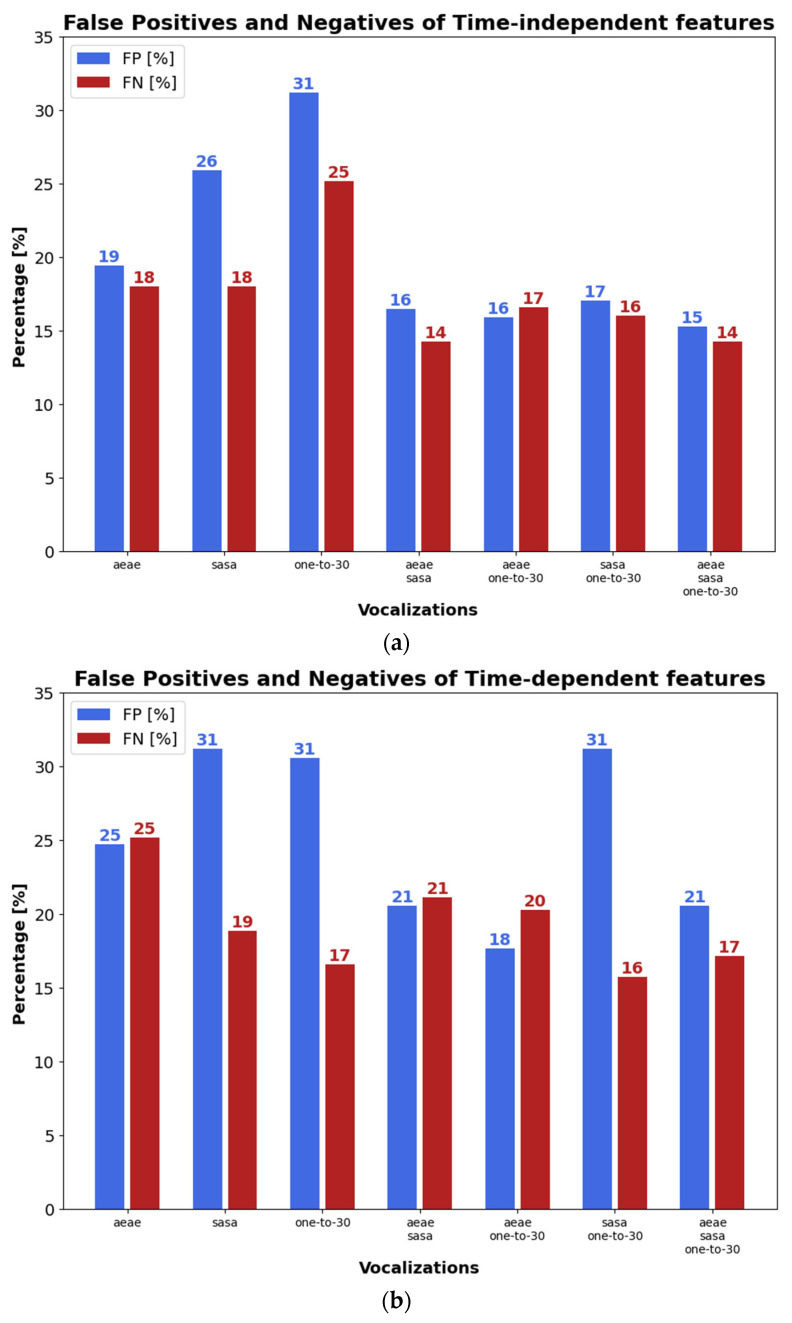
FP and FN with single and combined classifiers: (**a**) time-independent features; (**b**) time-dependent features; and, (**c**) combined time-independent and time-dependent features.

**Figure 7 sensors-23-02441-f007:**
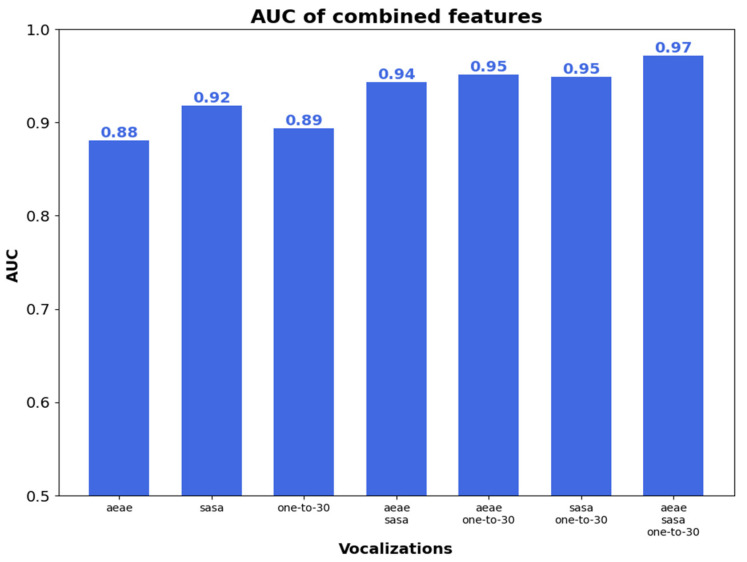
AUC with time-dependent and -independent features.

## Data Availability

Not applicable.

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
