# Peer review of "Dyspnea Severity Assessment Based on Vocalization Behavior with Deep Learning on the Telephone"

_sensors, 2023, doi:10.3390/s23052441_

Round 1
Reviewer 1 Report
Why is an automatic system necessary to assess the severity of dyspnea, if there is a validated instrument to do so that is also extremely easy to apply. It is necessary to justify why to apply intelligent audio analysis to this problem. What are the advantages of analyzing the audio instead of applying a questionnaire? What additional information can be found?
Why to build a classifier instead of a regressor. If the mMRC scale is numerical and indicates the level of severity (the higher the more severe), the most appropriate approach was to build a regressor.
Having built a classifier means that the reported results do not really reflect how good the system is to solve the proposed problem. The reported accuracy is very low. An accuracy of around 60% would not inspire confidence to use this tool. My opinion is that building a regressor and reporting the correlation and the error was the most appropriate.
It would be interesting to analyze the differences between the data obtained by telephone call and the Web page since the encoding, compression and filtering can be different in both cases. How this affects the estimation of mMRC should be discussed.
Author Response
Reviewer #1:
“Why is an automatic system necessary to assess the severity of dyspnea, if there is a validated instrument to do so that is also extremely easy to apply. It is necessary to justify why to apply intelligent audio analysis to this problem. What are the advantages of analyzing the audio instead of applying a questionnaire? What additional information can be found?”
The fact that the detection is by means of audio analysis makes it possible to standardize the evaluation, reducing variability or bias between different doctors who take a test like a questionnaire. In particular, although the dyspnea assessment questionnaire is easy for a doctor to apply, it is not for ordinary people, who may have difficulties in understanding or answering the questions, particularly in the case of elderly adults. Moreover, the response given in a questionnaire can be influenced by the patient's mood or habituation to the disease [1-3]. Also, there are many studies that focus on remote monitoring of respiratory diseases by means of audio analysis [4], indicating that it is a current topic in the state of the art. In addition to the importance of remote monitoring, automation allows greater scalability by not requiring a specialist evaluating each person and several automatic health applications have been proposed in the last few years. This discussion was incorporated in the revised version of the manuscript.
“Why to build a classifier instead of a regressor. If the mMRC scale is numerical and indicates the level of severity (the higher the more severe), the most appropriate approach was to build a regressor”
The classical classification loses the ordinality of the labels, since it considers these as independent [5]. However, the use of regression also suffers from the problem that the root mean square error assumes that the separation between adjacent levels of the mMRC scale would be uniform [6]. In fact, although not reported here, the regression performed worse overall than the classification-based system. It is important to emphasize that the regression restricts the flexibility in merging or combining the different modules to explore the complementarity of their outputs. For these reasons, neuron stick-breaking [7] was considered as a trade-off between both solutions, including ordinality in the classification problem. The stick-breaking layer provided better results in models of time-dependent features. This discussion was included in the revised version of the manuscript.
“Having built a classifier means that the reported results do not really reflect how good the system is to solve the proposed problem. The reported accuracy is very low. An accuracy of around 60% would not inspire confidence to use this tool. My opinion is that building a regressor and reporting the correlation and the error was the most appropriate.”
The reviewer should be aware that accuracy corresponds to estimate exactly the same reference mMRC scores. Despite the fact that 60% does not sound outstandingly high, is not low either considering that the achieved RMSE is equal to 0.98. This means that the classification error is in average between adjacent classes. In addition, the current manuscript addressed a task (i.e. assessing respiratory distress severity using the mMRC levels) that, to the best of our knowledge, had not been addressed before. Moreover, the discrimination between sick and healthy patients is high obtaining an AUC equal to 0.97. The comparison between classification and regression is included above.
“It would be interesting to analyze the differences between the data obtained by telephone call and the Web page since the encoding, compression and filtering can be different in both cases. How this affects the estimation of mMRC should be discussed.”
The developed platform allows using IVR phone calls or the web based application. However, the results presented in the manuscript concern the phonetizations obtained with IVR phone calls only. This is because most of the elder individuals that participated in the study were not familiar with or did not have smart phones. This issue was clarified in the current version of the manuscript. Nevertheless, despite the fact that the comparison suggested by the reviewer is interesting, be aware that multi-condition training based approaches do not distinguish how the cell phones are employed.
Reviewer 2 Report
The authors describe a system to assess dyspnea using Machine Learning, by utilizing the mMRC scale on the phone. The authors propose a method based on modelling the spontaneous behaviour of subjects while pronouncing controlled phonetizations.
This study found that the system provided an accuracy of 59% (i.e. estimating the correct mMRC), a root mean square error equal to 0.98, false positive rate of 6%, false negative rate of 11% and an area under the ROC curve equal to 0.97.
Although this study is interesting and the data from this study provides value to clinical practice, please elaborate on the strengths and limitations of this study.
How can this Machine learning be applicable to a patient with an acute infective phase?
The literature search to justify this study seems very superficial. Please add data/studies to support this study.
There are several grammatical mistakes throughout the manuscript that needs to be addressed.
Author Response
Reviewer #2:
“Although this study is interesting and the data from this study provides value to clinical practice, please elaborate on the strengths and limitations of this study.”
The authors would like to thank the reviewer for the comment. The system presented here is a candidate to be deployed in public health applications and can detect dyspnea automatically by making use of the telephone network with artificial intelligence without any prior knowledge or test on subjects. The application of this technology could help to monitor the population at risk of pneumonia by COVID-19 and to detect COVID-19 sequel. Also, it offers the opportunity to have a remote and reliable tool beyond the current pandemic. As well, it could be helpful to monitor respiratory diseases to evaluate the effect of pollution, to monitor pre-existing or occupational diseases such as byssinosis in bakers or those resulting from high-risk pulmonary tasks such as work in large-scale mining or firefighting with dangerous gasses, among other occupations with high respiratory compromise. Another strength is the scalability of the system, which allows the evaluation of dyspnea without the need for specialized personnel to take the test. A weakness of this study corresponds to the subjacent hypothesis that the manifestation of dyspnea does not depend on the illness that causes it. Discriminating dyspnea depending on its cause is out of the scope of this paper. However, the aforementioned hypothesis seems reasonable. The aim of the technology presented here was to detect dyspnea independently of the underlying cause. Nevertheless, to determine if there are differences in observed dyspnea with respect to gender, age, comorbidities, etc, can be proposed as future research. This discussion was incorporated in the current version of the manuscript.
“How can this Machine learning be applicable to a patient with an acute infective phase?”
Patients with severe respiratory distress should have no problem performing controlled phonetizations. In cases where the health status of the patients is so severe that it does not allow them to pronounce these controlled vocalizations (e.g. when connected to an artificial respirator), the proposed system is clearly not applicable. Nonetheless it could be used to detect worsening respiratory distress before reaching an acute phase. This discussion was incorporated in the revised version of the manuscript.
“The literature search to justify this study seems very superficial. Please add data/studies to support this study”
The authors believed that the literature search in the original version of the manuscript was complete enough for this kind of multidisciplinary manuscript. However, as suggested by the reviewer, the following discussion was incorporated in the revised version of the manuscript. Moreover, this issue was also addressed in the reply to previous comments.
In recent years there has been a considerable increase in the development of remote monitoring health tools, due to the growing demand for health services [8]. There has been a great effort to try to prevent COPD with the use of machine learning (ML), since these methods are effective in collecting and integrating a diversity of medical data on a large scale in precision medicine [9]. This development has been boosted by the COVID-19 pandemic [10], where several artificial intelligence (AI)-based solutions have been proposed to automatically detect SARS-CoV-2 [11-14]. These ML based solutions have been focused mainly on smart phones due to the great scalability, ubiquity, and flexibility that these devices offer [15]. Usually, these systems are useful for disease monitoring, preventing people from having to go to medical centers. For example, in [16] the respiration rate is remotely evaluated by using phone sensors. In [17] the importance of telemedicine for people with COPD is evaluated, and in [18] the advantages of remote monitoring for patients with interstitial lung diseases are studied. Generally, the relevance of remote monitoring for people with chronic critical illness who have already been discharged is discussed in [19].
“There are several grammatical mistakes throughout the manuscript that needs to be addressed.”
Thank for pointing out this issue. The manuscript was proofread by a professional service.
REFERENCES
[1] Stoeckel M. C., Esser R. W., Gamer M., Büchel C., and von Leupoldt, A. Brain mechanisms of
short-term habituation and sensitization toward dyspnea. Frontiers in psychology, 2015, 6, 748. https://doi.org/10.3389/fpsyg.2015.00748
[2] Wan L., Stans L., Bogaerts K., Decramer M., and Van den Bergh O. Sensitization in medically unexplained dyspnea: differential effects on intensity and unpleasantness. Chest, 2012, 141(4), 989–995. https://doi.org/10.1378/chest.11-1423
[3] Von Leupoldt A., Dahme B. Psychological aspects in the perception of dyspnea in obstructive pulmonary diseases. Respir Med. 2007, 101(3), 411-22. doi: 10.1016/j.rmed.2006.06.011. Epub 2006 Aug 8. PMID: 16899357.
[4] Xia T., Han J., Mascolo C. Exploring machine learning for audio-based respiratory condition screening: A concise review of databases, methods, and open issues. Exp Biol Med (Maywood). 2022, 247(22), 2053-2061.
[5] Frank E., Hall M. A Simple Approach to Ordinal Classification. Lecture Notes in Computer Science. 2001, 2167, 145-156. 10.1007/3-540-44795-4_13.
[6] Berg A., Oskarsson M., O'Connor M. Deep Ordinal Regression with Label Diversity. 2020 25th International Conference on Pattern Recognition (ICPR). 2021, 2740-2747. 10.1109/ICPR48806.2021.9412608.
[7] Liu X., Fan F., Kong L., Diao Z., Xie W., Lu J., You J. Unimodal Regularized Neuron Stick-breaking for Ordinal Classification. Neurocomputing. 2020, 388, 34-44. 10.1016/j.neucom.2020.01.025.
[8] Duggal R, Brindle I, Bagenal J. Digital healthcare: regulating the revolution. Br Med J, 2018, 360, k6.
[9] Feng Y., Wang Y., Zeng C., Mao H. Artificial Intelligence and Machine Learning in Chronic Airway Diseases: Focus on Asthma and Chronic Obstructive Pulmonary Disease. Int J Med Sci. 2021, 18(13), 2871-2889. doi: 10.7150/ijms.58191. PMID: 34220314; PMCID: PMC8241767.
[10] Veenis J.F., Radhoe S.P., Hooijmans P., Brugts J.J. Remote Monitoring in Chronic Heart Failure Patients: Is Non-Invasive Remote Monitoring the Way to Go? Sensors (Basel). 2021, 21(3), 887. doi: 10.3390/s21030887. PMID: 33525556; PMCID: PMC7865348.
[11] A. Shoeibi et al. Automated detection and forecasting of covid-19 using deep learning techniques: A review. arXiv 2020 preprint arXiv:2007.10785.
[12] M. Ghaderzadeh and F. Asadi. Deep learning in the detection and diagnosis of COVID-19 using radiology modalities: a systematic review. Journal of healthcare engineering 2021, vol. 2021.
[13] A. M. Rahmani et al. Automatic COVID-19 detection mechanisms and approaches from medical images: a systematic review. Multimedia Tools and Applications 2022, pp. 1–20.
[14] N. Subramanian, O. Elharrouss, S. Al-Maadeed, and M. Chowdhury. A review of deep learning-based detection methods for COVID19. Computers in Biology and Medicine. 2022, p. 105233.
[15] Amiriparian S, Schuller B. AI hears your health: computer audition for health monitoring. In: Proceedings of the conference on health and wellbeing, 2021, pp.227–33
[16] Valentine S., Cunningham A.C., Klasmer B., Dabbah M., Balabanovic M., Aral M., Vahdat D., Plans D.. Smartphone movement sensors for the remote monitoring of respiratory rates: Technical validation. Digit Health. 2022, 8, 20552076221089090. doi: 10.1177/20552076221089090. PMID: 35493956; PMCID: PMC9052820.
[17] Franek J. Home telehealth for patients with chronic obstructive pulmonary disease (COPD): an evidence-based analysis. Ont Health Technol Assess Ser. 2012, 12(11), 1-58. Epub 2012 Mar 1. PMID: 23074421; PMCID: PMC3384362
[18] Wijsenbeek M.S., Moor C.C., Johannson K.A., Jackson P.D., Khor Y.H., Kondoh Y., Rajan S.K., Tabaj G.C., Varela B.E., van der Wal P., van Zyl-Smit R.N., Kreuter M, Maher T.M. Home monitoring in interstitial lung diseases. Lancet Respir Med. 2023, 11(1), 97-110. doi: 10.1016/S2213-2600(22)00228-4. Epub 2022 Oct 4. PMID: 36206780.
[19] Viderman D., Seri E., Aubakirova M., Abdildin Y., Badenes R., Bilotta F. Remote Monitoring of Chronic Critically Ill Patients after Hospital Discharge: A Systematic Review. J Clin Med. 2022, 11(4), 1010. doi: 10.3390/jcm11041010. PMID: 35207287; PMCID: PMC8879658
Round 2
Reviewer 1 Report
My comments were satisfactorily addressed